# Extracellular Vesicles Released by Enterovirus-Infected EndoC-βH1 Cells Mediate Non-Lytic Viral Spread

**DOI:** 10.3390/microorganisms8111753

**Published:** 2020-11-08

**Authors:** Eitan Netanyah, Matteo Calafatti, Jeanette Arvastsson, Eduardo Cabrera-Rode, Corrado M. Cilio, Luis Sarmiento

**Affiliations:** 1Immunovirology Unit, Department of Clinical Sciences, Skåne University Hospital, Lund University, 221 85 Lund, Sweden; eitan.netanyah@med.lu.se (E.N.); matteocalafatti@gmail.com (M.C.); jeanette.arvastsson@med.lu.se (J.A.); 2Department of Immunology, National Institute of Endocrinology, Vedado 10400, Havana, Cuba; diabetes@infomed.sld.cu or

**Keywords:** enterovirus, extracellular vesicles, exosomes, beta cells, type 1 diabetes, virus spread

## Abstract

While human enteroviruses are generally regarded as a lytic virus, and persistent non-cytolytic enterovirus infection in pancreatic beta cells has been suspected of playing a role in type 1 diabetes pathogenesis. However, it is still unclear how enteroviruses could exit the pancreatic beta cell in a non-lytic manner. This study aimed to investigate the role of beta cell-derived extracellular vesicles (EVs) in the non-lytic enteroviral spread and infection. Size-exclusion chromatography and antibody-based immunoaffinity purification were used to isolate EVs from echovirus 16-infected human beta EndoC-βH1 cells. EVs were then characterized using transmission electron microscopy and Multiplex Bead-Based Flow Cytometry Assay. Virus production and release were quantified by 50% cell culture infectious dose (CCID_50_) assay and qRT-PCR. Our results showed that EVs from echovirus 16-infected EndoC-βH1 cells harbor infectious viruses and promote their spread during the pre-lytic phase of infection. Furthermore, the EVs-mediated infection was not inhibited by virus-specific neutralizing antibodies. In summary, this study demonstrated that enteroviruses could exit beta cells non-lytically within infectious EVs, thereby thwarting the access of neutralizing antibodies to viral particles. These data suggest that enterovirus transmission through EVs may contribute to viral dissemination and immune evasion in persistently infected beta cells.

## 1. Introduction

Human enteroviruses are non-enveloped viruses of 25 to 30 nm in diameter with a single positive-strand RNA genome belonging to the *Enterovirus* genus of the Picornaviridae family. Over 100 distinct human enteroviruses serotypes are currently recognized, which are grouped into 4 species (namely species *Enterovirus A–Enterovirus D*). Human enteroviruses typically cause acute and self-limiting diseases (such as meningitis, myocarditis, hand-foot-and-mouth disease, and respiratory illness) in which the virus appears to be eliminated by the host immune responses [1]. In addition to acute diseases, human enteroviruses have also been associated with chronic conditions, such as type 1 diabetes (T1D) [2].

Despite the mounting evidence that members of species *Enterovirus B* can infect the pancreatic islets containing beta cells, a long-term puzzle in the pancreatic islet field has been how the virus–cell interactions dictate the course of the beta cell dysfunction that characterizes T1D [3,4]. A common feature of enteroviruses is the rearrangement of the cytoplasm of infected cells and the recruitment of host factors on specific membrane sites in order to facilitate viral genome replication [5]. The new virions are then released from the cell by a lytic mechanism to infect neighboring cells and eventually cause extensive tissue damage [1]. Strikingly, such damage is not typically observed in islets of patients with T1D [6,7,8,9], so that a persistent (non-cytolytic) enteroviral infection capable of evading the host’s immune surveillance, rather than an acute lytic infection, is postulated to stand for the key factor responsible for the progressive loss of insulin-producing pancreatic beta cells [10]. Yet, it is still unclear how enteroviruses, typically considered cytolytic viruses, can establish such an infection.

Many cell types, including those of the pancreatic islets, release into the extracellular environment diverse types of membrane vesicles of endosomal (exosomes, 50–100 nm) and plasma membrane origin (microvesicles, 100–1000 nm) under physiological or pathological conditions [11,12]. As enteroviruses are obligate intracellular pathogens, it is not surprising that they have evolved strategies to hijack the host cell vesiculation machinery to their profit. In this context, non-lytic spread of virions hiding within extracellular vesicles (EVs) has emerged as an alternative means of intercellular transmission of viral populations, as it does not alarm the immune system [13]. It has been previously shown that carcinomic human cervix epithelial cells (HeLa cells) respond to encephalomyocarditis virus, a close relative of human enterovirus, by releasing multiple EVs during the pre-lytic phase of infection [14]. Notably, poliovirus type 1 Mahoney (a member of the species *Enterovirus C)* and coxsackievirus B3 (a member of the species *Enterovirus B)* can also exit HeLa cells non-lytically through secreted vesicles harboring large numbers of infectious particles, contributing to an enhancement of the virus cell-to-cell transmission [15,16]. Likewise, exosomes released from rhabdomyosarcoma cells infected with enterovirus 71 (a member of species *Enterovirus A*) contain infectious viruses that can establish a productive infection in human neuroblastoma cell lines [17]. Additionally, the hepatitis A virus (a member of the Picornaviridae family) can be released from cells cloaked in host-derived membranes that resemble exosomes [18].

Even though in vitro studies have shown that enteroviruses use EVs to escape host cells and facilitate viral spread, the mechanism of EVs-mediated transmission of enterovirus infection in pancreatic beta cells is still largely unknown. Thus, in the present study, we used the EndoC-βH1 immortalized human beta cell line as a model for human beta cells to investigate whether human enteroviruses exploit the EVs produced by beta cells to non-lytically reach the extracellular environment and maximize virus dissemination.

## 2. Material and Methods

### 2.1. Cell Culture

The human pancreatic beta cell line EndoC-βH1 (courtesy of Professor Erik Renstrom, Lund University) was grown in a culture plate coated with 2 μg/mL of fibronectin (Sigma-Aldrich, St Louis, MO, USA)/1% extracellular matrix (ECM, Sigma-Aldrich, St Louis, MO, USA) and cultured at 37 °C and 5% CO_2_ in low-glucose (1 g/L) Dulbecco’s Modified Eagle Medium (DMEM, Gibco, Thermo Fisher Scientific, Waltham, MA, USA) with 2% albumin from bovine serum fraction V (Roche Diagnostics), 50 µM 2-mercaptoethanol (Sigma-Aldrich, St Louis, MO, USA), 10 mM nicotinamide (Sigma-Aldrich, St Louis, MO, USA), 5.5 μg/mL transferrin (Sigma-Aldrich, St Louis, MO, USA), 6.7 ng/mL sodium selenite (Sigma-Aldrich), and penicillin (100 units/mL)/streptomycin (100 μg/mL) (Gibco, Thermo Fisher Scientific, Waltham, MA, USA).

### 2.2. Virus

The Echovirus 16 (E16, member of the species *Enterovirus B*) strain used in this study was originally isolated from the stool of a patient who developed T1D autoantibodies (i.e., islet cell antibodies [ICA], insulin autoantibodies [IAA], and glutamic acid decarboxylase antibodies [GADA]) [19]. Importantly, E16 can replicate in human primary pancreatic islets and rodent beta cell lines [20,21]. The identity of the isolate was confirmed by partial VP1 sequences by primer pairs 187 (VP1; 5′-ACIGCIGYIGARACIGGNCA-3′) and 011 (2A; 5′-GCICCIGAYTGITGICCRAA-3′) (Thermo Fisher Scientific Waltham, MA, USA) and comparison with published sequences. To generate viral stock, confluent green monkey kidney (GMK) cells were infected with E16 and incubated at 36 °C at 5% CO_2_ until cytopathic effect (CPE) was observed. Cell cultures were freeze-thawed three times to lyse all the cellular membranes, media was collected, and cell debris was removed by centrifugation at 400× *g* for 10 min. The titer of viral stock was determined using end-point dilutions in microwell cultures of GMK cells and expressed as a 50% cell culture infectious dose (CCID50) per mL according to the Spearman–Karber method [22].

### 2.3. Viral Replication

EndoC-βH1 cells were plated at 4 × 10^5^ mL^−1^ in a 24-well tissue culture plate and infected with E16 at the indicated multiplicity of infection (MOI) when they reached 80–90% confluence. Sets of plates corresponding to the number of time points were incubated with the inoculum at the same initial time, using a distinct flat-bottom 24 well-plates for each time point. After absorption for 2 h at 36 °C, cells were washed twice with phosphate-buffered saline (PBS) to remove any unattached virus and the 2 h time point plate was frozen to determine viral background levels. One mL of fresh DMEM medium (Gibco, Thermo Fisher Scientific, Waltham, MA, USA) was added to the cell culture; thereafter, cells and supernatants were harvested at the indicated time points following infection. Supernatants were cleared from cell debris by differential centrifugation for the determination of extracellular infectivity. Adherent cells were rinsed twice with PBS and frozen at −80 °C. Intracellular infectivity levels were assessed from the cell pellet after three consecutive freeze-thaw cycles to release all the intracellular viruses. The amount of infectious viral particle (CCID_50_) was determined both in the supernatant and cell pellet by using end-point dilutions in microwell cultures of GMK cells [22].

### 2.4. Cell Viability

The viability of EndoC-βH1 cells was determined by a modified 3-(4.5-dimethylthiazol-2-yl)-2.5-diphenyltetrazolium bromide (MTT) assay (Thermo Fisher Scientific, Waltham, MA, USA), which measures the mitochondrial metabolic rate. The measurement of lactate dehydrogenase (LDH) activity in the cell culture supernatant of EndoC-βH1 cells, which results from a loss of cell membrane integrity, was performed by using the lactate dehydrogenase (LDH) cytotoxicity assay kit (Thermo Fisher Scientific, Waltham, MA, USA) according to the manufacturer’s guidelines.

### 2.5. Isolation of EVs

EndoC-βH1 cells mock-infected or infected with E16 were cultured in a medium containing albumin (cleared from exosomes by ultracentrifugation for 16–20 h at 100,000× *g*). Culture supernatant was collected at 24 h post-infection (hpi) and centrifuged at 300× *g* for 10 min to remove cells and larger cell debris. The supernatant was then centrifuged at 2000× *g* for 10 min and filtered through a 0.22-μm filter (Millipore, Sartorius AG, Göttingen, Germany) to remove the cell debris further. After, 1 mL of the vesicle-containing supernatant was loaded onto commercially available qEV original size exclusion columns (iZON Science, Oxford, UK). The columns contain a resin with a pore size of approximately 70 nm and a bed volume of 10 mL. Fractions 7–9 (1.5 mL) were collected with each fraction being 0.5 mL. The purified EV preparations were used for immunoaffinity capture using beads coated with antibody specific for the EpCAM membrane antigen (EpCAM magnetic microbeads from Invitrogen, Thermo Fisher Scientific, Vilnius, Lithuania) according to the manufacturer’s protocol. The bead-bound EVs complexed were washed twice with PBS and EVs were eluted from the magnetic beads with 2.15 M NaCL in PBS and incubated overnight at 4 °C. Collected EVs were used for RNA extraction and inoculation in cell culture immediately after their isolation. Protein concentration was quantified using the BCA assay kit for low concentrations (Pierce Biotechnology, Rockford, IL, USA).

### 2.6. Transmission Electron Microscopy (TEM) 

Isolated vesicles (5 µL) were mounted on 400-mesh carbon-coated copper grids (EMS) and incubated in a humidifying chamber for 30 min at room temperature. After, the grid was transferred onto drops of 2% uranyl acetate (*w*/*v*) and washed several times with ultra-pure water for negative staining. Each carbon grid was vacuum evaporated and analyzed using Joel Model JEM-1230 transmission electron microscope operated at 80 kV.

### 2.7. Flow Cytometry of EVs

Purified EVs from cells were adsorbed on 4-μm streptavidin beads (Invitrogen Dynabeads) pre-coated with anti-CD63 by incubating 10 μL of beads with 5 μg of CD63 antibody from BioLegend (BioLegend, San Diego, CA, USA) overnight protected from the light on an orbital shaker (450 rpm) to allow complete bead saturation at 4 °C. EVs were washed with 1X PBS and centrifuged to remove unbound particles. The bound EVs were immunostained against exosome-associated proteins CD9, CD63, and CD81 (BioLegend, San Diego, CA, USA) and analyzed on a FACScalibur flow cytometer (BD Biosciences) using FlowJo software version 7.6.5 (TreeStar, Ashland, OR, USA). Corresponding isotype control IgGs were used as negative controls.

### 2.8. Multiplex Bead-Based Flow Cytometry Assay (MACSPlex)

EVs surface markers profiles were investigated with the MACSPlex exosome kit, human (Milteny Biotect, GmbH, Bergisch-Gladbach, Germany). MACSPlex Exosome Kits makes it attainable to detect 37 markers (CD1c, CD2, CD3, CD4, CD8, CD9, CD11c, CD14, CD19, CD20, CD24, CD25, CD29, CD31, CD40, CD41b, CD42a, CD44, CD45, CD49e, CD56, CD62P, CD63, CD69, CD81, CD86, CD105, CD146, CD209, CD326, CD133/1, CD142, MCSP, SSEA-4, ROR1, HLA-ABC, and HLA-DRDPDQ) plus two isotype controls (mIgG1 and REA) to determine the unspecific binding of the EVs. The product comprises a cocktail of various fluorescently labeled bead populations, each coated with a specific antibody binding the respective surface epitope. Isolated EVs (4–20 μg) were diluted with the provided MACSPlex buffer at a final volume of 120 μL in 1.5-mL tubes before 15 μL of MACSPlex Exosome Capture Beads were added to each tube. Tubes were then incubated on an orbital shaker overnight (14–16 h) at 450 rpm at room temperature protected from light. To wash the beads, 1 mL of the provided MACSPlex buffer was added to each tube and centrifuged at room temperature at 3000× *g* for 5 min. For counterstaining the EVs bound by capture beads with detection antibodies, 135 μL of the MACSPlex buffer and 5 μL of each APC-conjugated anti-CD9, anti-CD63, and anti-CD81 detection antibody were added to each tube and incubated on an orbital shaker at 450 rpm protected from light for 1 h at room temperature. The beads were washed twice with 1 mL of MACSPlex buffer and centrifuged at room temperature at 3000× *g* for 5 min. Subsequently, 150 μL of MACSPlex buffer were added to each tube, and beads were resuspended by pipetting up and down. Flow cytometric analysis was performed with a CytoFlex Flow Cytometer (Beckman Coulter, Inc. Miami, FL, USA), and data were analyzed using the CytExpert Software (Beckman Coulter, Inc. Miami, FL, USA). Median fluorescence intensity (MFI) values for all the capture bead subsets were corrected for background signal by subtracting the respective MFI values from matched non-EV buffers that were treated exactly like EVs-containing samples (buffer/medium + capture beads + antibodies). All results were normalized to combined mean MFI values of the surface proteins CD9, CD81, and CD63 to determine the relative levels of surface marker expression.

### 2.9. Enterovirus Genome Detection

The detection and quantification of the presence of enteroviral RNA was performed by using the commercial OneStep qRT-PCR MasterMix kit (PrimerDesign, UK) and Enterovirus genesig quantification kit (PrimerDesign, UK) containing primers/probe mix for enterovirus and internal control RNA as well as enterovirus -spp-positive control template for the standard curve. The primers and probe sequences in this kit have 100% homology with over 95% of the NCBI database sequences available from clinically relevant reference enterovirus. Viral RNA was extracted using the QIAmp viral RNA Mini kit according to the manufacturer’s instructions (QIAGEN GmbH, Hilden, Germany). The internal extraction control RNA provided by Primer Design was added to the RNA sample once it was resuspended in lysis buffer. This control RNA was then co-purified with the sample RNA and detected as a positive control for the extraction process. Each sample was run in triplicate and measured in the FAM- and VIC-channels on an ABI PRISM 7900 (Applied Biosystems ViiA™ Real Time PCR System, Life technologies, Foster city, CA, USA). The viral copy number was calculated using the standard curve method according to the manufacturer’s protocol. A validated RNA-positive extract from E16 stock was used as a positive control and RNA-free water was used as a negative control. Under optimal PCR conditions, the assay can detect between 1 × 10^3.3^ and 1 × 10^8.3^ copies of the target template per mL.

### 2.10. Virus Neutralization Assay

A specific neutralizing antibody against E16 was prepared by immunizing New Zealand white rabbits with antigens obtained from E16 stock. Eight to 10 mL of antigen was injected intravenously on days 0, 7, and 21. Heart puncture bleeding was induced 7 days after the last injection and the neutralization capacity of the E16 antisera was evaluated by microneutralization on 96-well cell culture plates with the use of serial dilutions, by a factor of 4, of serum beginning at 1:8. At each dilution, 25 μL of serum was mixed with 25 μL of Eagle’s medium containing 100 TCID50 (range between 32 to 320) of E16. The virus–serum mixture was incubated for 3 h at 37 °C in an atmosphere of 5% CO_2_. One hundred microliters of GMK cell suspension (2 × 10^5^ mL^−1^) were added and incubated for 5 days. The neutralizing antibody titer was calculated according to the Reed and Muench method and expressed as the reciprocal titer of the highest serum dilution factor that resulted in a 50% inhibition of the cytopathic effect [23]. For neutralization assay, the E16 antiserum was diluted to give a final concentration that prevents infection of 100% of replicate inoculations. Equal quantities of free virus (100 CCID_50_) or EVs were incubated with E16-specific neutralizing antibodies for 2 h at 37 °C in an atmosphere enriched with 5% CO_2._ The serum–virus mixture was added into pre-seeded EndoC-βH1 cells in a 24-well plate and incubated at 37 °C for 2 h. After adsorption, the inoculum was removed, the cells were washed with serum-free DMEM, and they were incubated at 37 °C for 96 h in a 5% CO_2_ incubator. Samples were tested in triplicate and each test batch was accompanied by a cell control, a “serum toxicity” control (for the possible cytopathic effect of the serum alone), virus dose, and titration controls. The amount of infectious viral particles (CCID_50_) was determined in total EndoC-βH1 cell lysates (cells plus supernatants) by using end-point dilutions in microwell cultures of GMK cells [22].

### 2.11. Statistical Analysis

Statistical analyses were performed using Prism software (with GraphPad Prism version 8.1). Experiments were performed a minimum of three times unless otherwise stated. Gene expression, viral titers, and viability assays were analyzed using unpaired Students *t*-test with Welch’s correction. Data are presented as means ± SEM and a *p*-value of <0.05 was considered significant in all experiments (* *p* < 0.05, ** *p* < 0.01, *** *p* < 0.001).

### 2.12. Ethics Statement

The study was conducted in compliance with the principles expressed in the Declaration of Helsinki and the European Council’s Convention on Human Rights and Biomedicine. All methods were carried out following relevant guidelines and regulations. All protocols and experimental procedures involving animals used in this study for the production of hyperimmune serum were reviewed and approved by the local ethics committee for animal experimentation at National Institute of Endocrinology under the permit number 0704030.

## 3. Results

### 3.1. E16 Egress from Infected EndoC-βH1 Cells before Lysis of the Cell’s Plasma Membrane

To investigate whether enteroviruses are released from beta cells in a non-lytic fashion, EndoC-βH1 cells were infected with E16 for 24, 48, and 72 h at different MOI, starting from an intended MOI = 1 and followed by 10-fold serial dilutions. Intracellular virus production and extracellular virus release along with the cell viability and plasma membrane integrity were monitored throughout infection. Time-course analysis of viral titers revealed that EndoC-βH1 cells were highly permissive and produced high titers of infectious virus and lytic infections by 24 h after virus challenge at an input MOI as low as 0.0001. These data collectively suggest that EndoC-βH1 cells are highly susceptible to E16 infection. Remarkably, extracellular viral production (1.99 log_10_ CCID_50_/mL increase above the background level) without perturbing cell’s membrane integrity or cell viability of EndoC-βH1 cells was observed by 24 hpi at an MOI of 0.00001 (Figure 1A–C). It is noteworthy that the increase of extracellular virus titers was at a lower level than those for intracellular infectious viruses (2.32 log_10_ CCID_50_/mL), suggesting the presence of a considerable number of viral particles in the intracellular compartments at this time point (Figure 1A). Thereafter, E16-infected cells displayed a significant decrease in cell viability (Figure 1B; *p* = 0.019 at 48 h, *p* = 0.0004 at 72 h) and plasma membrane integrity (Figure 1C; *p* = 0.0048 at 48 h, *p* = 0.0001 at 72 h). Although the possibility of individual cell lysis cannot be completely ruled out, these results suggest that, before cell lysis, E16 exits EndoC-βH1 cells via non-lytic pathways.

### 3.2. Enterovirus-Infected EndoC-βH1 Cells Release Extracellular Vesicles That Are Infectious and Mediate Non-Lytic Viral Spread

We next questioned whether E16 egressed from EndoC-βH1 cells in a non-lytic fashion within infectious EVs. Isolation of enterovirus-free EVs may be challenging since some subtypes of EVs and human enteroviruses display a similar size. To overcome this limitation, we used qEV, a size-exclusion chromatography (SEC) separation technology, to purify EVs from EndoC-βH1 cells that were mock infected or infected with E16 at an MOI of 0.00001 for 24 h. Using an isolation strategy based on SEC qEV columns, EVs with the size range between 70 and 250 nm normally elute largely in fractions 7–9, while proteins, free E16 virus, and other contaminating molecules smaller than 70 nm elute in later fractions. Indeed, cup-shaped vesicles with a morphology and size compatible with exosome (70–100 nm) (Figure 2A and Appendix A) and microvesicle size (100–200 nm) (Figure 2B and Appendix A) were identified in fractions 7–9 by TEM. Remarkably, the TEM analysis yielded images showing virus-like particles within enclosed membrane structures (Figure 2C), which are similar to 28–30 nm E16 particles that eluted in fraction 16–18 (Figure 2D).

Flow cytometry analysis demonstrated the presence of the most conspicuous exosome markers (i.e., CD9, CD63, and CD81) in EVs purified by SEC, validating successful isolation of EVs from EndoC-βH1 cells (Figure 3A and Appendix A). Further phenotyping analysis using MACSPlex showed that both mock-EVs and E16-EVs were enriched with cell-to-extracellular matrix adhesion molecules (CD29), cell-to-cell adhesion molecules (CD24, CD44, and CD326), tissue factor/CD142 protein, transmembrane glycoprotein CD133/1, HLA ABC, and EVs marker proteins (CD9, CD63, and CD81). The remaining markers (27 out of 37) had a lower MFI than those from the isotype controls, indicating a lower expression. Remarkably, EVs from EndoC-βH1 cells exhibited strong signals for CD326 (EpCAM) compared with other EV surface markers and its expression was increased by 1.8 fold post-infection (Figure 3B and Appendix A).

To further exclude cell-free virions in the EVs preparations, SEC-isolated EVs were then subjected to immunomagnetic selection with EpCAM antibody-coupled magnetic beads. Applying this strategy to the EVs isolated from E16-infected cells, we evidenced the presence of enteroviral RNA in EpCAM-positive EVs (Figure 4A). We subsequently performed a simulation experiment to validate this result where we mixed cell-free E16 with control EVs isolated by the SEC qEV columns and then further purified by EpCAM-based immunomagnetic capture. As expected, we did not detect enteroviral RNA in the mock-infected EVs exposed to free E16. In contrast, viral RNA was recovered in the flow-through after magnetic separation and removal of supernatant (Figure 4B). These findings confirmed that EVs isolation using SEC qEV columns followed by EpCAM-based purification results in a pure population of EVs that are not contaminated with viruses. We then tested whether total EpCAM selected EVs from viral-infected cells were infectious. To this end, EVs isolated by the SEC qEV columns from E16-infected EndoC-βH1 cells or mock-infected cells were purified using anti-EpCAM immunoaffinity beads and then tested for infectivity on naive EndoC-βH1 cells. In naïve EndoC-βH1 cells infected with EpCAM-positive EVs from E16-infected cells, the results revealed that viral titers increased above background levels within 48 h of infection in a time-dependent manner (Figure 4C). No increase in viral titers above background levels was detected in EndoC-βH1 cells treated with EpCAM-positive EVs from mock-infected cells. Altogether, these results suggest that E16 exits EndoC-βH1 cells non-lytically via enclosure in EVs that can shuttle virus to naïve beta cells and establish a productive infection.

### 3.3. E16-Containing Infectious EVs Promote Viral Spreading Even in the Presence of Neutralizing Antibodies 

Given that vesicles containing E16 from EndoC-βH1 cells were infectious, we next investigated whether the presence of neutralizing antibodies can block EVs-mediated intercellular transmission of E16. To test this, we produced a polyclonal rabbit antiserum against E16 and the virus-neutralizing capacity was determined by calculating the highest dilution of serum that gave 100% virus neutralization. The calculated neutralization titer against 100 CCID_50_ of E16 was 1:8192 (Appendix A). Once we quantified the virus-neutralizing capacity of this polyclonal antibody, we tested the virus-neutralizing activity on EVs-mediated infectivity. The titration of viral production showed that EVs-mediated infectivity was largely unaffected when E16-specific neutralizing antibodies were incubated with EVs from E16-infected EndoC-βH1 cells. In contrast, incubation of free virus with the same amount of antibody resulted in a significant reduction of the viral production, with titers not exceeding the background after 96 hpi (Figure 5). These results indicate that EVs-mediated E16 transmission is not inhibited by virus-specific neutralizing antibodies.

## 4. Discussion

A long-held tenet of virology holds that non-enveloped RNA viruses exit cells via lysis of the plasma membrane [1]. Over recent years, this paradigm has changed and there is now mounting evidence that some non-enveloped viruses, including enteroviruses, can exit cells via non-lytic pathways [24]. We have previously demonstrated that most intracellular enteroviral populations are released from virus-infected beta cell line INS (832/13) prior to cell lysis [25]. Still, the precise mode of non-lytic viral release from beta cells remains obscure. The main finding of this study is that enteroviruses can exit infected beta cells in EVs before cell lysis occurs. We further showed that EVs-mediated enteroviral infection of beta cells is resistant to antibody neutralization, suggesting that enteroviral transmission through EVs might be a viral strategy to escape from the immune detection.

Our analysis of surface signatures of EVs derived from mock and E16-infected EndoC-βH1 cells revealed the presence of proteins widely used as “markers” of EVs, such as CD9, CD63, and CD81. This is consistent with the view that human pancreatic islets and beta cell-derived lines can secrete EVs in a constitutive and stimulus-dependent manner [26,27,28]. We also found that EpCAM, a transmembrane glycoprotein primarily known to mediate cell contacts in epithelial tissues [29], is highly expressed in EndoC-βH1 cells-derived EVs. Of note, EpCAM is known to be associated with exosomes through the interaction with different tetraspanin proteins, such as the CD9 and CD44 variant isoform [30]. Interestingly, studies have indicated strong EpCAM expression in fetal pancreatic islet cells in situ and after induction of cell growth in vitro [31]. It is therefore conceivable that EVs from EndoC-βH1 cells were enriched in EpCAM. Since EpCAM plays a pivotal role in cell-to-cell and cell-to-extracellular environment interactions, it is tempting to speculate that increased EVs expression of EpCAM in E16-infected EndoC-βH1 cells could be involved in EV uptake by naïve beta cells. However, the biological role of EpCAM expression by beta cell EVs was not assessed here. To our knowledge, this hypothesis has not been previously described. Therefore, further studies are required to gain more comprehensive knowledge about the possible role of EpCAM in EV-mediated transmission of enterovirus infection among beta cells. Whether the results obtained with EndoC-βH1 cell-derived EVs can be extrapolated to other beta cell lines or isolated islets also needs to be addressed.

Given that we observed a marked expression of EpCAM in EndoC-βH1 cell-derived EVs, we further purified EpCAM-positive EVs from the total EVs isolated via SEC. By using this approach, we ruled out the presence of cell-free viral particles in the EV preparation, thereby further underscoring the ability of EVs derived from E16-infected beta cells to spread the infection successfully. Indeed, simulation experiments confirmed that there were no free E16 viral particles in the purified EpCAM-positive EVs, providing further evidence that viral particles are associated with EndoC-βH1 cell-derived EVs rather than contamination with cell-free virus.

Electron microscopic analysis revealed the heterogeneity in the EVs’ size, where virus-like particles could be observed inside the EVs, especially in those that correspond to intermediate-sized EVs. This finding is consistent with the earlier observation that enterovirus-infected HeLa cells release infectious virus during the pre-lytic phase of infection in multiple types of EVs that differ in size and molecular composition [14]. Because our analysis is restricted to EVs that range in size between 70 and 250 nm, a limitation of our study is that EVs above the size of 250 nm will be under-represented, making it unclear to what extent large EVs actually contribute to the non-lytic viral release and spread from infected beta cells. In fact, it has been shown that Coxsackievirus B3 can egress from cells non-lytically in vesicles larger than 500 nm derived from secretory autophagosomes [16].

While the biogenesis route, mechanisms of cargo recruitment, and mechanisms of release of EVs are not yet fully established, dysregulation of autophagy is increasingly viewed as one of the potential mechanisms that can influence cargo sorting and release of EVs [32,33,34]. Our previous work showed that E16 impairs autophagy in human pancreatic islet cells, resulting in a net accumulation of cytoplasmic autophagosomes as a consequence of impaired delivery of these vesicles to the lysosomal compartments for degradation [25]. Furthermore, it has been shown that autophagosomal membranes capture viral capsids at a time coinciding with a decrease in viral RNA synthesis [35]. Since secretory autophagy is an alternative disposal strategy under conditions of impaired autophagic degradation, it is reasonable to predict that enterovirus-infected beta cells would seek to rid themselves of virus-capturing vesicles, either via fusion with endosomal multivesicular bodies and subsequent release of exosome-enclosed virus particles or via fusion with the plasma membrane and subsequent release of microvesicle-like EVs carrying enteroviral particles.

The absence of extensive beta cell death despite the detection of enterovirus RNA and protein in pancreas biopsies of T1D patients has led to speculation that persistent viral infection (perhaps in a slowly replicating form) might play a role in T1D development [36]. In order to persist, enteroviruses must avoid the clearance mediated by the immune system and minimize host cell lysis, which would otherwise leave the virions susceptible to be neutralized by antibodies. A key finding reported in this study is that EVs-mediated transmission of E16 in EndoC-βH1 cells is resistant to antibody neutralization. Thus, it could be argued that the pre-lytic release of E16 in EVs shields the virions from host neutralizing antibodies or patrolling immune cells and thereby might counteract viral clearance.

In summary, this study demonstrates for the first time that enterovirus-infected beta cells secrete infectious EVs before the lytic release of virions, which can mediate virus transmission in a neutralizing antibody-resistant manner. Thus, the present study is starting to shed light on the mechanisms by which enterovirus escape immune surveillance and establish persistent beta cell infections.

## Figures and Tables

**Figure 1 microorganisms-08-01753-f001:**
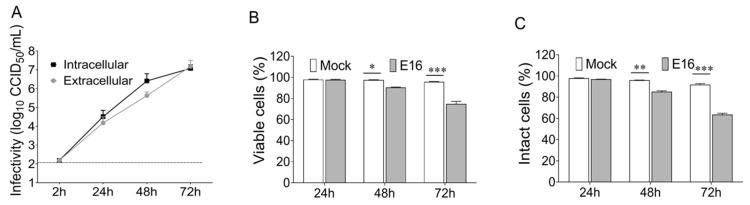
Pre-lytic release of echovirus 16 (E16) in EndoC-βH1 cells. (**A**) Cells and corresponding culture supernatant were harvested at indicated time points after infection of EndoC-βH1 cells with E16 (MOI = 0.00001) and the intracellular virus production and extracellular virus release was determined by 50% cell culture infectious dose (CCID50) assay in green monkey kidney (GMK) cells. Dotted black lines indicate the limit of detection. (**B**) Cell viability measured by 3-(4.5-dimethylthiazol-2-yl)-2.5-diphenyltetrazolium bromide (MTT) assay and (**C**) plasma membrane integrity of EndoC-βH1 cells by determining lactate dehydrogenase (LDH) activity in culture supernatant at the indicated time points after E16 infection (MOI = 0.00001) compared to mock-treated cells. Data are presented as mean ± SEM for three independent experiments, with each measurement performed in triplicate. * *p* < 0.05, ** *p* < 0.01, *** *p* < 0.001.

**Figure 2 microorganisms-08-01753-f002:**
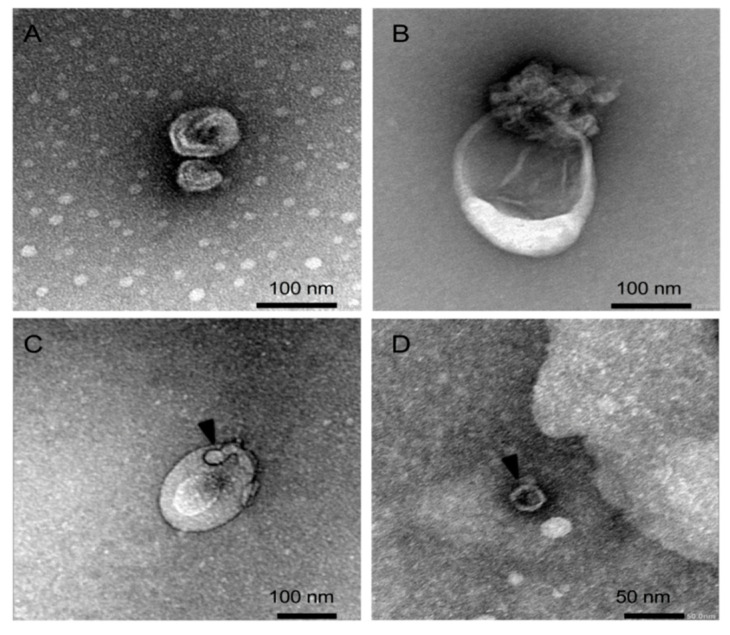
Representative transmission electron microscopy image of negatively stained EVs isolated from (**A**–**C**) E16-infected EndoC-βH1 cells (fraction 7-9) and (**D**) viral stock solution (fraction 16–18) using SEC qEV columns. Black arrowheads indicate viral particles. See Appendix A for further details.

**Figure 3 microorganisms-08-01753-f003:**
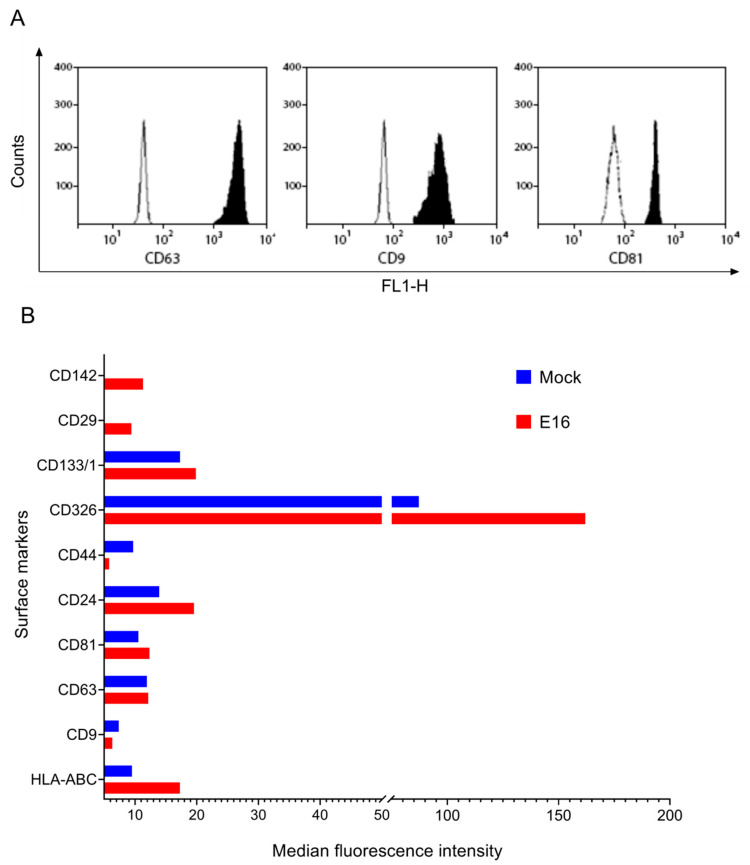
Phenotyping of EndoC-βH1 cell-derived EVs (**A**) Flow cytometer results showing the expression levels of CD63, CD9, and CD81 in EndoC-βH1 cell-derived EVs. Barren areas are isotype control and shaded areas represent positive marker expression. See Appendix A for complete datasets of all replicates (**B**) Median fluorescence intensities values of selected markers detected in EVs isolated from mock- and E16-infected EndoC-βH1 cells. The figure shows representative data of at least five independent experiments using bead-based multiplex flow cytometry assay. See Appendix A for complete datasets of all replicates.

**Figure 4 microorganisms-08-01753-f004:**
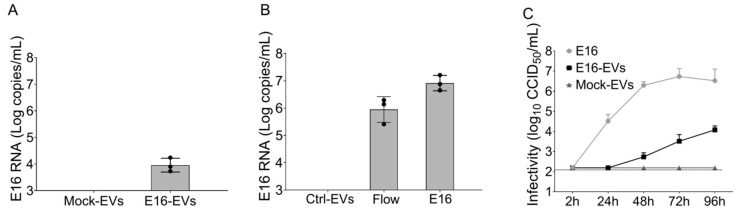
EVs released from E16-infected EndoC-βH1 cells transmit productive enterovirus infection to recipient cells. (**A**) Real-time qPCR analysis of enterovirus RNA content in immunopurified EpCAM-positive EVs from mock (Mock-EVs) or E16-infected EndoC-βH1 cells (E16-EVs). (**B**) EVs derived from mock-infected EndoC-βH1 cells were exposed to free E16 and re-isolated by the immunomagnetic selection of EpCAM-positive EVs. Enterovirus RNA content was analyzed by qPCR in EpCAM-selected EVs (Control-EVs) and flow-through samples. Cell-free virus (E16) was used as a positive control. (**C**) Naïve EndoC-βH1 cells were incubated with EVs from E16-infected EndoC-βH1 cells (E16-EVs) or mock-infected cells (Mock-EVs). EndoC-βH1 cells infected with the cell-free virus (E16) at an MOI of 0.00001 were used as a positive control. At 2 hpi, cells were washed off, and cultured in fresh Dulbecco’s Modified Eagle Medium (DMEM) medium for further 96 h. Total EndoC-βH1 cell lysates (cells plus supernatants) were collected at different time points and viral titers determined by end-point dilution on GMK cells. Dotted black lines indicate the limit of detection. Data are representative of three independent experiments, with each measurement performed in triplicate (mean ± SEM).

**Figure 5 microorganisms-08-01753-f005:**
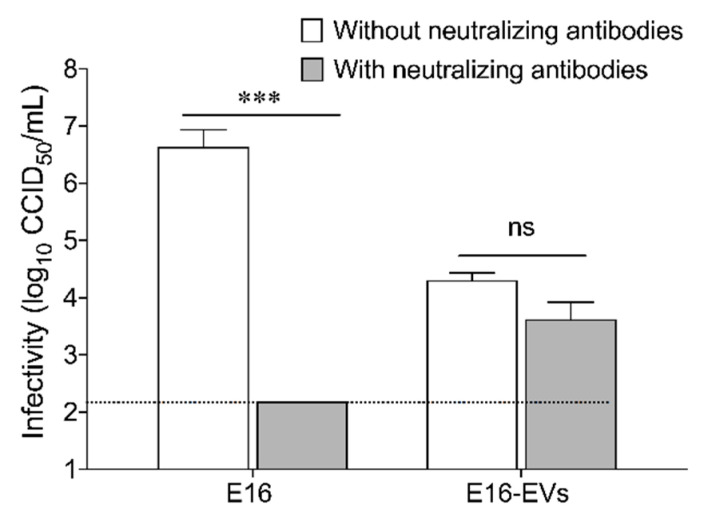
EVs-mediated E16 infection of EndoC-βH1 cells is not inhibited by virus-specific neutralizing antibodies. Free virus (E16) and EVs from E16-infected EndoC-βH1 cells (E16-EVs) were incubated with or without anti-E16 neutralizing antibodies. Titration of viral production in the presence or absence of neutralizing antibodies was determined as described in the materials and methods. Dotted black lines indicate the limit of detection. Data are representative of three independent experiments, with each measurement performed in triplicate (mean ± SEM). *** *p* < 0.001.

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
