# Peer review of "Extracellular Vesicles Released by Enterovirus-Infected EndoC-βH1 Cells Mediate Non-Lytic Viral Spread"

_microorganisms, 2020, doi:10.3390/microorganisms8111753_

Round 1
Reviewer 1 Report
In this study, Netanyah et al investigate the possibility that pre-lytic vesicles released from E6 infected beta cells are capable to mediate further infection. To do so they assessed cell integrity at different timepoints after initial infection, isolated the putative pre-lytic vesicles via SEC and immunoisolation against EpCam that is normally not present in cell-free virus. They found viral RNA in EPCAM positive samples and showed that those particles are infectious but less efficiently than classical post-lytic virus. They conclude that pre-lytic vesicles confer infectivity.
Distinction/relationship between EVs and “pure” virus is the focus of intense research effort and discussions (Nolte-'t Hoen E, Cremer T, Gallo RC, Margolis LB. Extracellular vesicles and viruses: Are they close relatives? Proc Natl Acad Sci U S A.. doi: 10.1073). The possibility that pre-lytic EVs emanating from infected cells are capable to confer infectivity is interesting.
Unfortunately, the data generated in this interesting study are too preliminary to support the authors claim
.
I have Two major concerns:
- It is possible that the method used to detect cell lysis is not sensible enough. Indeed I totally agree with the authors who carefully conclude line 142 “the possibility of individual cell lysis cannot be completely ruled out”. This prevent therefore to formally conclude on non-lytic but potent vesicles.
- Strict separation of Viruses from EVs is not trivial and may be virtually impossible. Although the authors claim that SEC is an efficient method to separate both type of particles, I would like to highlight that, to my knowledge, there is no rigorous proof of such a separation. In addition, although the authors tried to use immuno-isolation to remove any possible contamination with cell-free virus that are normally depleted of EpCam marker, it is still possible that viral RNA that is detected from EPCAM positive vesicles is indeed emanating from virus that are docked to the EV surface and that are co-isolated. This again prevent to rigorously conclude that the sample is free of “classical virus”.
Other Comments:
Figure 2 shows only few vesicles, which prevents to convincingly demonstrate that vesicles found in each fraction are really different. Furthermore, Size is only one parameter that is not sufficient to discriminate particles types.
Figure 4: It is not clear if the vesicles (E6-EV) loaded on target cell for infectivity assessment were
- first immuno-isolated (against EpCam) and then tested for infectivity on target cell,
or
- Collected from conditioned media at early timepoint prior detectable cell lysis of donor cell, without further immuno-isolation.
In the former case (1), I do not understand how vesicles can be released from the beads without disruption, prior to being functionally tested for infectivity. In the latter case (2), how can the authors rule out the presence of classical E6 virus in the sample then?
Author Response
Thank you very much for your time reviewing our manuscript “Extracellular Vesicles Released by Enterovirus-Infected EndoC-βH1 Cells Mediate Non-Lytic Viral Spread” (Manuscript ID: microorganisms-977091)”. We highly appreciate your constructive suggestions and comments, which substantially contributed to our manuscript. Please find attached our responses (in red) to the reviewers’ comments.

Reviewer 2 Report
The manuscript submitted by Netanyah et al. is of interest and relevance. However, some major points need to be addressed.
1) The abbreviation EVs (for extracellular vesicles) is misleading as it is often used for enteroviruses.
2) Introduction section: Line 68: Not all Coxsackieviruses belong to Enterovirus Species B.
3) Material and Methods section: From my point of view, a permit in terms of animal welfare is required for the production of the antisera. Please include the number of the application and the corresponding approval authority.
4) Results:
- Figure 1: The used MOI of 0.00001 is rather low. It means one infectious particle per 100,000 cells. What was the reason to use such a low MOI? Did you observe differences when higher MOIs were used (compare line 236, you started with a MOI of 1)? This also applies to the other experiments.
- Figure 2: It's not entirely clear if some of the electron microscope images could also be artifacts. In this context, an illustration should be added to the supplement containing several examples per illustration part.
- Figure 3: In the supplement it should be proven that the figure part A is a representative figure from several independent experiments. Part B of the figure lacks the mean values and standard deviations.
- Figure 4: Is the RT-PCR used for the detection of E16 plus-strand specific? Have the results been normalized to the added standard? Could one try to compare the levels of cellular RNA in the different extractions?
- Line 337: The NT should be demonstrated in the supplement.
Author Response

(The authors gave the same response as above.)

Reviewer 3 Report
This is an interesting study that focus on the mechanisms underlying persistent enterovirus infections using a pancreatic beta cell line. The manuscript is well written and the experimental sections are consistent with the proposed hypothesis.
Specific comments:
1. Did the authors try or think about using a different pancreatic cell line for comparison to EndoC-βH1? Can these results be extrapolated to other beta cell lines or to isolated islets?
2. It is mentioned that EpCAM might play an important role in cell-to-cell and cell-to-extracellular environment interactions. The expression of EpCAM in infected and non-infected cells could be easily studied by IHC or IF together with exosome markers or even with VP1. The same experiment can be done in EndoC-βH1 after infection with E16-EVs. The content of the manuscript could therefore be improved and the hypothesis of the possible involvement of EpCAM in the uptake of this sort of E16-extracellular vesicles could be experimentally confirmed.
Author Response

(The authors gave the same response as above.)

Round 2
Reviewer 2 Report
Most of the experts' recommendations have been adequately taken into account. For my part, I just want to make two comments:
1) The supplementary figure 1 contains only four examples. In my opinion, it would be better to give more examples.
2) In the supplementary figure 3, the scaling of the abscissa is not uniform. This does not make it clear how variable the results actually are.